# Efficacy of Submicron Dispersible Free Phytosterols on Non-Alcoholic Fatty Liver Disease: A Pilot Study

**DOI:** 10.3390/jcm12030979

**Published:** 2023-01-27

**Authors:** María C. Brañes, Raimundo Gillet, Rodrigo Valenzuela

**Affiliations:** 1Naturalis Research Consortium, Santiago 8700548, Chile; 2Department of Nutrition, Faculty of Medicine, University of Chile, Santiago 8380000, Chile

**Keywords:** liver steatosis, plant sterols, water-dispersible phytosterols, metabolic parameters

## Abstract

Background: No pharmacological treatment is yet approved for non-alcoholic fatty liver disease (NAFLD). Plant sterols have shown healthy properties beyond lowering LDL-cholesterol, including lowering triglycerides and lipoprotein plasma levels. Despite pre-clinical data suggesting their involvement in liver fat control, no clinical study has yet been successful. Aims: Testing a sub-micron, free, phytosterol dispersion efficacy on NAFLD. Methods: A prospective, uncontrolled pilot study was carried out on 26 patients with ≥17.4% liver steatosis quantified by magnetic resonance imaging. Subjects consumed daily a sub-micron dispersion providing 2 g of phytosterols. Liver fat, plasma lipids, lipoproteins, liver enzymes, glycemia, insulinemia, phytosterols, liposoluble vitamins and C-reactive protein were assessed at baseline and after one year of treatment. Results: Liver steatosis relative change was −19%, and 27% of patients reduced liver fat by more than 30%. Statistically and clinically significant improvements in plasma triglycerides, HDL-C, VLDL and HDL particle number and C-reactive protein were obtained, despite the rise of aspartate aminotransferase, glycemia and insulinemia. Though phytosterol plasma levels were raised by >30%, no adverse effects were presented, and even vitamin D increased by 23%. Conclusions: Our results are the first evidence in humans of the efficacy of submicron dispersible phytosterols for the treatment of liver steatosis, dyslipidemia and inflammatory status in NAFLD.

## 1. Introduction

Non-alcoholic fatty liver disease (NAFLD) is the most frequent liver pathology in occidental societies, with a prevalence of 25% in general population [1]. In people consuming less than 30 g of alcohol per day, NAFLD corresponds to the accumulation of triglycerides (TG) in more than 5% of hepatocytes with no inflammation or fibrosis, as demonstrated by liver biopsy. Subsequently, it can progress to non-alcoholic steatohepatitis (NASH), which includes liver inflammation without or with fibrosis. If the latter is the case, it can later turn into cirrhosis. Approximately 25% of patients may reach this stage and experience its complications, including hepatocellular carcinoma and liver transplant. Consistently, it has been reported that in NAFLD, an increment in oxidative stress parameters and the pro-inflammatory status of the liver may be metabolic conditions related to the progression towards other more aggressive liver injuries [2].

Currently, there is no specific treatment for NAFLD. The therapeutic approach has thus far been concerned with delaying or reversing the disease progress by treating concomitant pathologies such as, insulin resistance with sensitizers, dyslipidemia with lipid control using atorvastatin or ezetimibe, and obesity with bariatric surgery [3]. Novel pharmaceutical approaches targeting farnesoid X receptors have been shown to reduce hepatic fibrosis and inflammation through regulation of lipid and glucose metabolism [4]. The most advanced drug, Ocaliva, has presented unwanted side effects, such as increased low-density lipoprotein-cholesterol (LDL-C) levels and pruritus. Pharmacological approaches based on one compound have not been successful in more than 30% of NASH or fibrosis patients, and high doses present adverse effects [3]. Therefore, combined approaches are a plausible strategy in the future.

Plant sterols have been demonstrated to have a role in human metabolism for many decades. Currently, 2 g of phytosterols are recommended for the management of dyslipidemia and prevention of cardiovascular (CV) disease [5]. Phytosterols’ regulatory activity on plasma LDL-C levels has been explained through increased fecal neutral sterol loss by inhibition of intestinal cholesterol absorption [6,7], and trans intestinal cholesterol flux [8]. More recent findings have shown reduction in plasma TG, especially in subjects with high baseline levels [9,10], and the regulation of apoprotein B (Apo B) metabolism [11]. Lately, pre-clinical data have also demonstrated the positive influence of phytosterols in the gut microbiome [12]. These suggest that additional as yet unravelled mechanisms must be involved. Therefore, a plausible related regulation could occur through mechanisms controlling liver lipids, which would thus have a role in NAFLD alleviation.

As free plant sterols are solid lipids that melt over 140 °C, they precipitate in different food matrices. The food industry has partially overcome this difficulty by decreasing their melting temperature to 60 °C through esterification with a fatty acid, improving their incorporation into lipid food matrices. Though pre-clinical data suggest that plant sterols reduce liver fat content [13,14] and inflammation [15], clinical studies have failed to show their efficacy in NAFLD [16,17,18,19,20]. We have used a submicron, water-dispersible free phytosterol (SDP) suspension. This particular formulation allows avoiding fatty foods as matrices. Clinically, in mildly hypercholesterolemic subjects, SDP has not only shown LDL-C-lowering efficacy, but also and unexpectedly, it has shown TG-lowering efficacy, even in a group of non-hyper triglyceridemic subjects [21]. In accordance with this background, we proposed to study the beneficial effects of administering this SDP daily in a one-year, prospective open pilot study with a cohort having a moderate to severe liver fat content (≥17.4%) together with either dyslipidemia, altered hepatic enzymes or ultra-sensitive C-reactive protein (us-CRP), which could relate to a NASH state.

## 2. Materials and Methods

The study protocol was designed and carried out by the Clinical Trials and Medical Research Study Center (Santiago, Chile) according to the ethical guidelines of version 7 of the Declaration of Helsinki [22], as reflected in a priori approval by the Central Ethics Committee of the Metropolitan Chilean Ministry of Health, filed under No. 474/2017. All volunteers gave written informed consent before participating in the study.

### 2.1. Study Design

This was an open-label, interventional, uncontrolled pilot study. Subjects were asked to maintain their lifestyle and feeding habits and to consume SDP daily for one year with one glass of water before lunch. Audit and Short Form-36 Health Survey questionnaires were answered at the start and end of the study. Additionally, lifestyle habits were controlled in visits every 3 months and in monthly telephone calls. The main aim was to evaluate the safety and efficacy of the treatment with SDP for liver fat reduction, and specifically to determine if a clinically significant reduction in liver fat (≥30% relative to baseline) [23] and improvements in liver enzymes, TG and us-CRP levels occurred. The secondary outcome was a clinically significant improvement in anthropometric measurements, glycemia control-related parameters, arterial pressure, lipid profile and body fat.

### 2.2. Inclusion Criteria

Adults between 18 and 90 years old with a body fat determined by dual absorptiometry X-ray of >25% (men) and >30% (women), and an ultrasonography diagnosis of moderate liver steatosis and NAFLD diagnosis with liver fat average ≥17.4%, confirmed by magnetic resonance imaging proton density fat fraction (MRI-PDFF), were included. Additionally, one of the following criteria should be met: TG > 150 mg/dL and <499 mg/dL; total cholesterol > 200 mg/dL; type 2 diabetes (T2D) and/or arterial hypertension; us-CRP > 1 mg/L; aspartate- (GOT) or alanine- (GPT) aminotransferases > twice the normal values, or NAFLD diagnosis during the last 6 months. 

### 2.3. Exclusion Criteria

The exclusion criteria were as follows: body mass index (BMI) < 17 kg/m^2^ or >40 kg/m^2^, liver diseases including cirrhosis, autoimmune hepatitis, hepatitis B or C, alcoholic fatty liver, hemochromatosis, cholangitis, any type of cancer, Wilson disease, sitosterolemia, pregnancy or breastfeeding women, alcohol consumption > 210 g/week for men and >180 g/week for women or AUDIT test points > 7, fasting glycemia > 126 mg/dL, GOT > 300 U/L, plasma creatinine > 2 mg/dL, parenteral nutrition the year before, previous bariatric surgery or under evaluation for one, biliary diversion, carrying a pacemaker or devices incompatible with MRI analysis; taking any plant sterols or treatment with thiazolidinediones, vitamins E, A, D and K, metformin, UDCA, SAM-e, betaine, silymarin, fibrates, ezetimibe, lipase inhibitors, resins, anti TNF-α therapies, probiotics, fish oil or omega-3 supplements, amiodaron, methotrexate, 3 months previous to this study. Other anti-diabetic treatments or statins were accepted if permanently taken 8 weeks before the screening visit.

### 2.4. Clinical and Metabolic Assessment

Blood biochemical analyses were carried out in Holland Laboratory (Santiago, Chile) according to standard clinical laboratory procedures. Liposcale^®^, serum lipoprotein profile characterization including lipid content (cholesterol and triglyceride concentration), size and particle number of the main lipoprotein classes was performed by Biosfer Teslab (Reus, Spain). Serum samples were shipped on dry ice and were kept at −80 °C until analysis. Serum (200 μL) was diluted with deuterated water (50 µL) and 50 mM phosphate-buffered solution at pH 7.4 (300 µL). Diffusion-ordered ^1^H NMR spectra were recorded at 306 K on a Bruker Avance III 600 spectrometer operating at a proton frequency of 600.20 MHz (14.1 T), as described in Mallol et al., 2015 [24]. Plasma sterols were quantified by gas chromatography, as described in Ahmida et al. [25]. Briefly, 200 μL of plasma was mixed with 200 μL of toluene solution containing 0.8 μg cholestane (97% purity; Sigma Aldrich) and were saponified at 70 °C for 1 h in a 1 M KOH/Ethanol solution, then extracted twice with a Hexane/Ethanol 20:1 *v*/*v* solution and derivatized with 200 μL BSTFA (99% purity; Sigma Aldrich) and 100 μL pyridine. The chromatographic analysis was performed using the Agilent HP-5MS 30 m × 0.25 mm × 0.25 μm column (Code 19091S-433) preheating at 90 °C for 3 min, a temperature ramp to 260 °C at 25 °C/min for 28 min, and a temperature ramp to 275 °C at 1 °C/min for 13 min. Injector and transference line temperatures were set at 270 and 230 °C, respectively, with splitless injection. Helium was used as a carrier gas with a flux of 1 mL/min and an injection volume of 1 μL.

### 2.5. Liver Steatosis Quantification

MRI-PDFF was performed in UC Christus Health Center (Santiago, Chile) using a Phillips Achieva 1.5 T instrument (Phillips, Best, The Netherlands) as described in Herrera et al. [26]. Briefly, PDFF was obtained using six echoes’ gradients and a repetition time/echo time/time between echoes of 30/1.3/2.1 msec. Twenty transversal cuts, 10 mm width each, were used to scan the whole liver. In every Couinaud liver segment, a region of interest was selected for obtaining the average of fat fraction. The steatosis degree according to Tang et al. [27] was used, where the following ranges are defined: normal ≤ 6.4% fat, low > 6.4%; moderate ≥ 17.4% and severe ≥ 22.1% fat.

### 2.6. Investigational Product (IP)

SDP was a bench-scale dispersion manufactured by Nutrartis LLC, Santiago, Chile. Each dose (10.5 mL) provides 2 g non-GMO, pine-derived free phytosterols (70% sitosterol, 15% campesterol, 15% sitostanol, 5% stigmasterol) dispersed in water using 1.5% food grade surfactants (polysorbate 80 and oleic acid salts). It is a submicron aqueous dispersion with a density between 1 and 1.01 g/mL and a pH between 8.5 and 10.5. Its particle size distribution, determined by dynamic light scattering, gave an average size of 300 nm, and percentiles 50, 75 and 90 corresponding to 270, 370 and 500 nm, respectively (Appendix A).

### 2.7. Statistical Analyses

After a descriptive analysis, the normal distribution of variables was evaluated using the Shapiro–Wilk test. Values correspond to median, 25th and 75th percentile (p25-p75). Assessment of significant differences for comparing the paired volunteer’s baseline and endpoint values was performed by a Wilcoxon statistical test. Statistical significance was set at an alpha level of 5%. For all analyses, the statistical software used was SPSS v.24.0 (Chicago, IL, USA).

## 3. Results

### 3.1. Subjects’ Characteristics

One hundred and sixty-eight volunteers were screened; 132 did not meet the inclusion criteria, and 36 were recruited from July 2017 until January 2020. Ten volunteers abandoned (7 because of adherence below 70% to investigational product (IP) consumption, 1 did not get his MRI-PDFF in the final visit, 1 had a surgical process unrelated to IP, 1 discontinued because of long-term hospitalization after infection with SARS-CoV-2). The general variables of participants are detailed in Table 1, and baseline conditions are detailed in Table 2, Table 3 and Table 4. The participants were 65% females, over 48 years old. Independent of gender, subjects were mainly obese, with more than 75% having a BMI over 30 kg/m^2^ and body fat exceeding 30%. Around half of them had a moderate level of liver steatosis (≥17.4% liver fat) and the other half had a severe level of steatosis (>22.1%). Predictive scores for fibrosis were used for excluding advanced fibrosis, as they have shown a good negative predictive value (NPV) [28]. Except for the BARD score, the other four scores had 0% and an average 85% for positive and NPV, respectively (Appendix A), indicating with high probability that subjects did not have a fibrotic liver. us-CRP suggested an incipient inflammatory state, and their lipid profiles denoted moderate hypercholesterolemia and hypertriglyceridemia. Nevertheless, liver enzymes, glycemia-related parameters and blood pressure were within normal values. Audit punctuation below eight indicated that fatty liver was not associated with alcohol consumption. Moreover, liver steatosis and elevated us-CRP in addition to dyslipidemia and overweight indicated that not only a NAFL, but possibly a NASH cohort had been recruited.

### 3.2. Liver and Plasma Lipid Level Comparison after Treatment with SDP

During the complete year protocol, no adverse effects related to IP consumption were reported. A trend of 19% relative reduction in liver fat was detected (Figure 1, Table 2). In Figure 2, individual liver fat changes between the baseline and end point show that nine patients had increased liver fat while 17 improved it to different degrees. Most importantly, almost one third experienced a clinically significant result of ≥30% relative reduction in MRI-PDFF. Median relative change in liver fat was associated with statistically significant improvements in parameters of the lipid profile, a 15% reduction in the median plasma TG value and a 7% increase in HDL-C. TG were reduced in all plasma lipoprotein fractions, but only the 19% decrease in VLDL-TG resulted in a statistically significant restoration of its median to normal values (<98 mg/dL). Total and LDL-C medians had minor variations and remained over normal ranges, while for VLDL- and IDL-C median values, despite showing relevant reductions, their median did not reach normal values of <22 and <9 mg/dL, respectively.

### 3.3. Plasma Concentration of Apoproteins, Lipoproteins and the Size of Lipoproteins after Treatment with SDP

Improvements in liver and plasma lipids detected after one-year treatment with SDP also included levels of lipoproteins and apoproteins (Table 3). Apo B levels, although in their normal range at baseline (40–145 mg/dL), showed a statistically significant reduction that was associated mainly with median VLDL-P decrease to normal levels (<70 nmol/L, (Figure 1), as LDL-P change was negligible. VLDL-P reduction corresponded with statistically significant decreases to normal values in M- and S-VLDL-P medians (<7.51 and <61 nmol/L, respectively). On the other hand, baseline and endpoint Apo A medians remained within the normal range (98–166 mg/dL) after a 4% trend to increase. Meanwhile, a statistically significant 7% increase in total HDL-P level was detected, mainly caused by an upgrade of p25 value which surpassed the normal lower limit of 24 nmol/L (Figure 1). The latter correlated with the 5% increase in S-HDL-P concentration that was detected. The size (Z) of the three lipoproteins was minimally affected by the treatment, and they did not reach the least clinically significant absolute difference value of 1 nm [29]. Nevertheless, 50% of participants improved their LDL-Z, as their median reached the expected value associated with low CV risk (>20.91 nm). VLDL- and HDL-Z remained within normal baseline ranges (42.03–42.36 nm for VLDL-Z and >8.21 nm for HDL-Z). Consistent with the improvement in liver steatosis, plasma lipids and lipoprotein levels, a statistically significant reduction in the inflammatory marker us-CRP was obtained (Table 4, Figure 1).

### 3.4. Biochemical, Anthropometric and Metabolic Parameters after Treatment with SDP

Despite the aforementioned improvements, IP efficacy was not exerted on other parameters, e.g., liver enzyme levels, anthropometric measurements and insulin resistance parameters (Table 4). All fibrosis scores were increased (Appendix A) in accordance with their dependence on these parameters, especially the BARD-score [28]. There were no differences in alcohol consumption and quality of life measures detected by Audit and SF36 questionnaires, respectively (data not shown). In control visits, patients suggested no major changes to their lifestyle. A statistically significant increase in baseline normal levels of the liver enzyme GOT lacked clinical significance (i.e., more than twice the highest normal value of 40 IU/mL). Moreover, the GOT/GPT ratio remained around normal values (<2). Most unexpectedly, improvements in lipid parameters occurred despite the clinically insignificant rise in glycemia, insulin and HOMA (Table 4).

SDP’s formulation is of pine tree origin, and its main composition is based on sistosterol (70%) and campesterol (15% of phytosterols). Accordingly, after the intervention, their plasma levels increased by 36 and 6%, respectively (Table 4). The increase in plasma phytosterols did not affect levels of carotenes, as they showed no significant variation and remained under normal range values (5–300 μg/dL). On the other hand, almost 50% of this cohort presented baseline insufficiency levels of vitamin D (<20 ng/mL). Unexpectedly, this parameter showed more than a 23% increase, affecting mainly subjects with the lowest baseline values (Table 4, Figure 1). Thus, the long-term consumption of SDP would not limit the availability of fat-soluble vitamins. An additional beneficial effect of SDP detected in this cohort was a slight but statistically significant decrease in diastolic blood pressure (Table 4, Figure 1).

## 4. Discussion

The present study is the first evidence of NAFLD improvement with long-term administration of free phytosterols to patients with moderate to severe steatosis, together with the restoration of various plasma markers for lipids, lipoproteins and inflammation and without unwanted side effects. Improvements were observed despite subjects maintaining their basal low grade physical activity and general lifestyle conditions.

Despite preclinical data strongly suggesting that plant sterols could have a role in fatty liver treatment [13,14,30], previous attempts to find efficacy of plant sterols for NAFLD treatment have been unsuccessful in reducing liver steatosis [16,17,19]. Similarly to other plant sterol trials in NAFLD cohorts [16,19], we also obtained a reduction in inflammatory biomarkers, in agreement with an anti-inflammatory role previously suggested for plant sterols [31] in obesity [20], post-menopausal states [32], and specifically in liver tissue [15].

One main difference between this study and previous ones has been the use of SDP. This formulation delivers nanoparticles with emulsified free phytosterols which have previously shown enhanced efficacy in lowering serum TG and the ratio of total to HDL cholesterol [21,33]. Here, additionally, liver fat reduction was observed. The mechanism explaining this remains to be elucidated. One hypothesis is based on the increased bioavailability of SDP in enterocytes, which could trigger the secretion of secondary metabolites relevant for targeting the metabolic pathways of lipid homeostasis. Another consequence of the increased bioavailability of SDP could be that higher levels of phytosterols may be transported to the liver through chylomicrons. The measurement of serum phytosterol concentrations after ingestion instead of fasting and steady-state levels of phytosterols would clarify this possibility, as the major sterol levels found in the plasma of fasting subjects were comparable to those found earlier [21] or provided by others [34,35]. Besides the potentially better functionality of the IP, the longer period of time of treatment in this trial might be an additional factor to consider. Other NAFLD protocols have tested 1 [16], 2 [19] and 4 [17] months of treatment. Few reports have shown the effects of consuming margarines enriched with sterol and stanol esters for more than a year, either in free living conditions [34] or clinically tested [35,36,37]. Therefore, this trial would be the first testing free phytosterols for the duration of one year and in a NAFLD cohort.

The 19% result obtained in liver fat reduction may reflect the fact that a subtle efficacy might require a high threshold in liver fat to obtain higher quantifiable results. Our protocol included an almost equal number of patients having a moderate and severe degree of liver fat determined by MRI-PDFF; this is a very objective, well-accepted technique for quantifying fat because it is noninvasive, accurate and reflects the state of the whole liver [38]. Others have used techniques recognized as non-quantitative, subjective methods, e.g., ultrasound imaging [16], frequently overestimating liver fat level [39], or have used liver enzymes as a surrogate marker of NAFLD alleviation [17]. Another hypothesis for the attenuated efficacy of the phytosterol treatment was the existence of non-responder subjects, presenting a high level of body cholesterol synthesis and a low level of gut absorption as a result of heterogeneity in genes participating in cholesterol metabolism [40]. We have no record of them in this trial, which could explain the null change in the LDL-C levels obtained. Other clinical trials have also used MRI-PDFF for testing the efficacy of new entity molecules [41,42] and feeding habits modifications [43,44] on liver steatosis. Despite different protocols and baseline conditions of subjects, the relative liver fat reduction and the proportion of subjects with a clinically relevant improvement in liver steatosis in these trials suggest that SDP could also form part of strategies to be considered after bigger and placebo-controlled trials confirm our preliminary findings.

The metabolic alterations of the studied cohort might be an important issue to consider for the efficacy of a functional ingredient on a specific parameter. Although there is an evident bidirectional link between NAFLD and metabolic syndrome (MS), disconnections between liver fat and insulin control have been reported [45] specifically in PNPLA3 gene variants, which are characteristic of the Hispanic population here studied, with susceptibility to NAFLD [46,47]. This fact could explain the different result obtained for waist circumference in MS subjects [33] and in this NAFLD cohort despite using the same IP. However, still, SDP unexpectedly also lowered TG in subjects not characterized by high TG values at baseline [21] as well as in MS subjects [33]. Plat et al. [48] had first hypothesized that the effect of stanols on TG could only be achieved if elevated baseline TG levels were present. It can alternatively or additionally be a matter of better bioavailability of SDP compared to ester formulations. Here, we have shown in NAFLD subjects that SDP lowered TG, as previously reported [49], and this occurred through a decreased VLDL-P output from the liver, consistent with the Apo B reduction also detected. High TG levels in all major Apo B-containing lipoprotein subclasses have been positively associated with higher risks of myocardial infarction [50], therefore highlighting the beneficial aspects that SDP may exert. Decreased Apo B and small dense LDL levels after administration of plant sterols and stanols have been described in hypercholesterolemic [51,52,53] and MS subjects [54].

The increase in HDL-C detected with this protocol overcame that obtained in mildly hypercholesterolemic subjects with the same functional ingredient as IP [21]. In addition, we detected that HDL-P levels were also increased, particularly from the small size fraction. Currently, HDL functionality in cholesterol efflux is still a matter of study [55]. Clinical evidence indicates HDL-P strongly and inversely correlates with incident CV disease, and thus has been suggested as a biomarker for prediction of CV events and residual risk [56]. Similarly, in non-pharmacologically treated populations, HDL-C would act as a protector while HDL-TG could also serve as a risk marker [50]. It remains to be understood whether the improvement in number is also related to HDL quality, as conflicting results depending on the method or the population context define large or small HDL particles as predictors of CV diseases [50,57].

There is concern about a possible loss of fat-soluble vitamins with phytosterol consumption [58]. Nevertheless, no significant reduction in carotenes was detected in this nor in previous studies with shorter periods of SDP administration [21,33]. Unexpectedly, vitamin D levels were significantly increased. Vitamin D and its receptor deficiencies have been associated with liver damage [59], inflammation [60] and fibrosis [61]. Cholecalciferol is transformed in the liver into 25-hydroxy-cholecalciferol [62] determined in plasma. Therefore, any condition improving liver function, reducing liver steatosis and thus inflammation might influence its metabolic performance, including 25-hydroxy-cholecalciferol production.

The effect of phytosterols on blood pressure has generally been a secondary outcome in some clinical trials [63]. The only report in a hypertensive cohort provided a mix as IP that impedes discrimination of the efficacy of phytosterols alone [64]. The supplementation of phytosterol esters for a one-year period to hypercholesterolemic healthy subjects showed no effect [35]. So, ours would be the first evidence of phytosterols’ involvement in blood pressure control. Head-to-head trials comparing SDP with sterol esters would clarify this issue.

In this NAFLD cohort, SDP consumption was safe during long-term use and showed a role in the control of plasma and liver lipids, but not in liver enzymes or controlled pro insulin resistance. SDP attenuated liver steatosis but did not reach the clinically relevant value of 30% expected in subjects that have a NAFLD diagnosis [23,38]. Therefore, it might be a valuable complementary therapeutic ingredient for current insulin sensitizers or other targeted therapies for treating NAFLD, besides being considered a prevention tool in overweight and obese subjects. The data presented, though coming from a small pilot trial, suggest that bigger placebo-controlled studies are required and worthy.

## 5. Conclusions

This is the first prospective clinical trial with plant sterols reducing liver fat. In addition, the lipid profile was optimized, particularly in TG and HDL-C and their respective apoprotein levels, but not in LDL-C. Other relevant results were the reduction of us-CRP and the recovery of deficient vitamin D levels. However, the treatment did not reduce the pathological progression of liver enzymes, glycemia and insulinemia, probably because subjects maintained their lifestyle habits during the year of intervention. Since no adverse events were reported under the prolonged administration of SDP, it might be considered an additional or complementary element as part of a preventive or therapeutic strategy for NAFLD. Undoubtedly, more robust clinical evidence would clarify our observations.

## Figures and Tables

**Figure 1 jcm-12-00979-f001:**
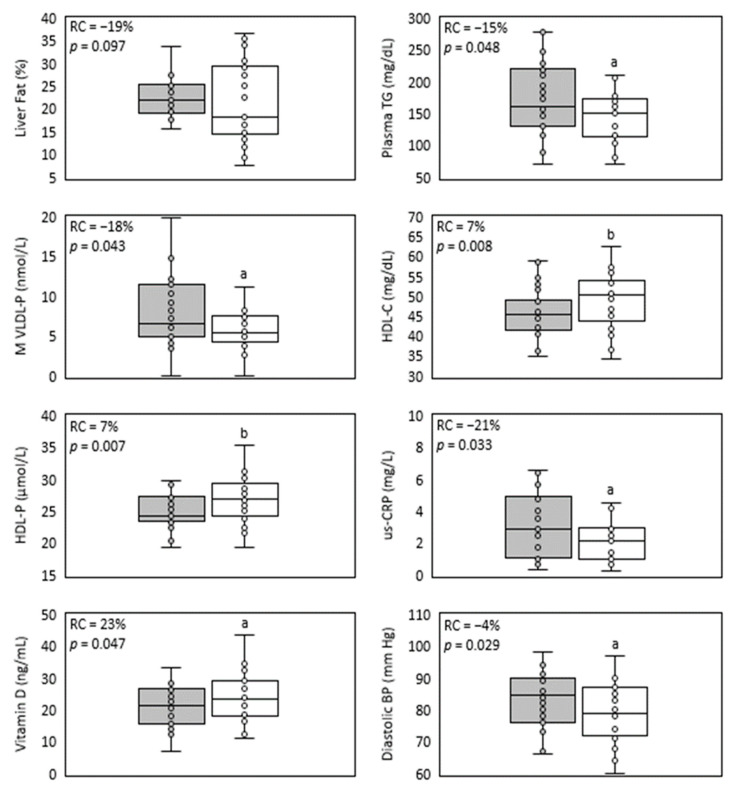
Main parameters improved in NAFLD by treatment with sub-micron phytosterols. Box and whisker plots of liver fat, plasma triglycerides, medium VLDL-particles, HDL-C, HDL-particles, us-CRP, vitamin D and diastolic blood pressure. The baseline values’ distribution is represented in the grey box, and the end values’ distribution is represented in the white box. The relative change median (RC) and *p*-value (*p*) are shown inside each plot. *n* = 26, except for lipids, lipoproteins and us-CRP (*n* = 25). a = *p* < 0.05 and b = *p* < 0.01 according to Wilcoxon test.

**Figure 2 jcm-12-00979-f002:**
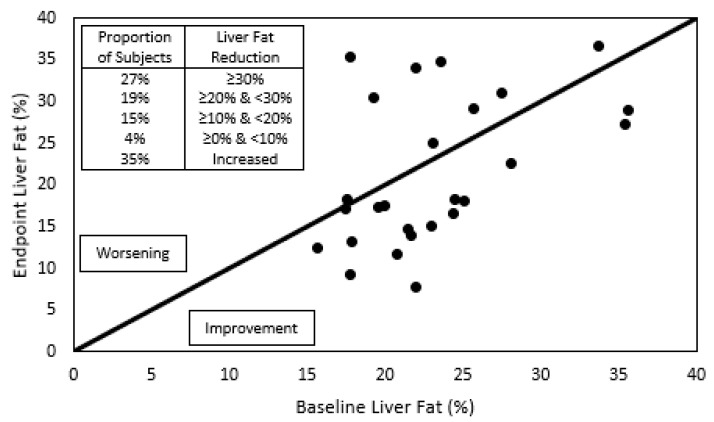
Individual liver fat content before and after phytosterol treatment. To the right of the diagonal line are located individuals whose liver fat improved after the one-year treatment with submicron dispersible phytosterols, and to the left, those who worsened. The insert indicates the proportion of subjects that showed a reduction in liver fat greater than 30%, between 30 and 20%, between 20 and 10%, between 10 and 0%, or those who showed an increase in liver fat.

**Table 1 jcm-12-00979-t001:** Sociodemographic variables and general health-related information of participants.

Variable	Median	p25	p75
Sociodemographic variables			
Age	54	48	59
Male	9	-	-
Female	17	-	-
General health-related information			
HBP	8	-	-
T2D	1	-	-
Alcohol consumption (yes)	17	-	-
Audit points	1.5	0.0	3.8

HBP and T2D: hypertension and type 2 diabetes under treatment. p25 and p75: 25th and 75th percentile. *n* = 26.

**Table 2 jcm-12-00979-t002:** Liver and plasma lipid levels’ comparison between baseline and after one year of treatment with sub-micron dispersible phytosterols.

Variable	Baseline	End	Relative Change	*p*-Value
Liver fat (%)	22.0 (19.4; 25.0)	18.2 (14.8; 29.1)	−19%	0.0972
TG (mg/dL)	160.0 (130.3; 215.9)	144.5 ^a^ (110.9; 172.8)	−15%	0.0478
VLDL-TG (mg/dL)	105.2 (85.2; 156.7)	93.8 ^a^ (67.9; 114.4)	−19%	0.0454
IDL-TG (mg/dL)	14.0 (11.8; 15.4)	12.7 (11.1; 14.9)	−4%	0.0853
LDL-TG (mg/dL)	21.5 (18.8; 23.5)	20.1 (17.6; 22.9)	−1%	0.2254
HDL-TG (mg/dL)	18.6 (14.7; 23.0)	18.5 (15.3; 22.2)	0%	0.2581
Tot-C (mg/dL)	232.5 (215.3; 248.5)	239.0 (216.6; 251.6)	1%	0.4051
VLDL-C (mg/dL)	27.0 (19.9; 36.7)	23.4 (17.9; 29.8)	−17%	0.0815
IDL-C (mg/dL)	14.2 (12.1; 16.1)	13.1 (10.2; 15.3)	−7%	0.0977
LDL-C (mg/dL)	147.5 (129.3; 153.9)	148.6 (131.0; 168.7)	0%	0.4002
HDL-C (mg/dL)	45.0 (41.4; 48.2)	49.8 ^b^ (43.3; 53.5)	7%	0.0079

Data include median values for each parameter and 25th and 75th percentile values in parentheses. Relative change is the median obtained from individual changes relative to baseline. *n* = 25, except in liver fat (*n* = 26). ^a^
*p* < 0.05, ^b^
*p* < 0.01, according to the Wilcoxon test.

**Table 3 jcm-12-00979-t003:** Comparison of plasma concentration of apoproteins, lipoproteins and the size of lipoproteins at baseline and after one year of treatment with sub-micron dispersible phytosterols.

Variable	Baseline	End	Relative Change	*p*-Value
Apo-B (mg/dL)	127.0 (113.0; 147.0)	117.0 ^b^ (107.2; 130.2)	−7%	0.005
VLDL-P (nmol/L)	83.3 (61.7; 108.7)	68.3 (50.8; 84.9)	−15%	0.050
L VLDL-P (nmol/L)	2.0 (1.6; 2.5)	1.7 ^a^ (1.4; 2.0)	−14%	0.037
M VLDL-P (nmol/L)	6.6 (5.0; 11.2)	5.4 ^a^ (4.5; 7.4)	−18%	0.043
S VLDL-P (nmol/L)	72.4 (56.1; 94.7)	60.1 ^a^ (44.2; 76.4)	−14%	0.045
LDL-P (nmol/L)	1517.1 (1409.6; 1749.8)	1601.5 (1348.6; 1702.9)	−1%	0.452
L LDL-P (nmol/L)	202.0 (175.6; 225.9)	198.6 (170.4; 218.1)	0%	0.340
M LDL-P (nmol/L)	432.7 (387.7; 502.5)	433.5 (362.1; 499.3)	−5%	0.293
S LDL-P (nmol/L)	882.2 (822.8; 1009.8)	892.7 (787.0; 993.7)	−1%	0.431
Apo-A (mg/dL)	128.0 (115.0; 147.0)	133.0 (122.1; 140.0)	4%	0.063
HDL-P (nmol/L)	24.3 (23.3; 27.0)	26.9 ^b^ (24.3; 28.3)	7%	0.007
L HDL-P (nmol/L)	0.3 (0.3; 0.3)	0.3 (0.3; 0.3)	0%	0.428
M HDL-P (nmol/L)	9.2 (8.7; 9.9)	9.9 (8.7; 10.7)	3%	0.089
S HDL-P (nmol/L)	15.0 (13.7; 16.5)	16.5 ^a^ (13.9; 18.3)	5%	0.043
VLDL-Z (nm)	42.0 (41.6; 42.1)	42.0 (41.9; 42.2)	0%	0.072
LDL-Z (nm)	20.9 (20.7; 21.0)	21.0 (20.8; 21.1)	0%	0.065
HLDL-Z (nm)	8.3 (8.3; 8.4)	8.3 (8.3; 8.4)	0%	0.331

Data include median values for each parameter and 25th and 75th percentile values in parentheses. Relative change is the median obtained from individual changes relative to baseline. L (large), M (medium), S (small), Z (particle size). *n* = 25. ^a^
*p* < 0.05; ^b^
*p* < 0.01 according to the Wilcoxon test.

**Table 4 jcm-12-00979-t004:** Diverse biochemical, anthropometric and vital signs parameters comparisons at baseline and after one year of treatment with sub-micron dispersible phytosterols.

Variable	Baseline	End	Relative Change	*p*-Value
us-CRP (mg/L)	3.2 (1.6; 5.0)	2.2 ^a^ (1.0; 3.5)	−21%	0.0331
GOT (U/L)	30.0 (23.3; 36.8)	33.0 ^a^ (29.0; 39.8)	21%	0.0322
GPT (U/L)	37.0 (30.0; 51.0)	36.0 (30.3; 56.0)	1%	0.4002
GOT/GPT	0.8 (0.6; 0.9)	0.8 (0.6; 1.0)	8%	0.0518
GGT (U/L)	29.5 (23.3; 37.0)	34.0 (22.8; 49.8)	3%	0.1372
Glycemia (mg/dL)	94.0 (86.0; 100.2)	96.0 ^a^ (90.5; 104.5)	7%	0.0232
Insulin (μUI/mL)	13.0 (8.7; 19.6)	14.7 (11.5; 22.0)	16%	0.0654
HOMA	3.0 (1.9; 4.4)	3.4 ^a^ (3.0; 5.0)	21%	0.0317
HbA1c (%)	5.6 (5.4; 5.9)	5.7 (5.4; 5.9)	1%	0.0537
BMI (kg/m^2^)	32.8 (30.5; 35.0)	32.4 (30.5; 34.6)	0%	0.3748
Waist circumf. (cm)	103.3 (98.3; 106.8)	103.0 (97.7; 109.3)	−1%	0.4576
Body fat (%)	42.6 (37.8; 46.4)	43.8 (37.2; 47.5)	0%	0.2223
Sistolic BP (mm Hg)	124.5 (118.0; 129.5)	126.5 (113.8; 130.8)	−2%	0.1383
Diastolic BP (mm Hg)	84.5 (76.5; 89.8)	79.0 ^a^ (72.5; 86.5)	−4%	0.0290
Vitamin D (ng/mL)	21.3 (16.3; 26.1)	23.2 ^a^ (18.6; 28.8)	23%	0.0471
Carotenes (μg/dL)	179.5 (133.3; 218.0)	165.0 (122.3; 206.5)	−7%	0.0952
Campesterol (μM)	8.4 (5.0; 11.1)	9.8 (7.2; 11.6)	6%	0.1423
Sitosterol (μM)	5.6 (4.0; 7.2)	6.4 ^a^ (5.3; 8.7)	36%	0.0173

Data include median values for each parameter and 25th and 75th percentile values in parentheses. Relative change is the median obtained from individual changes relative to baseline. *n* = 26, except in us-CRP (*n* = 25), body fat (*n* = 25), campesterol (*n* = 20), and sitosterol (*n* = 20). ^a^
*p* < 0.05 according to the Wilcoxon test.

## Data Availability

The data supporting the results of this study are all included in the present article.

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
