# Peer review of "Efficacy of Submicron Dispersible Free Phytosterols on Non-Alcoholic Fatty Liver Disease: A Pilot Study"

_jcm, 2023, doi:10.3390/jcm12030979_

Round 1

Reviewer 1 Report

The paper entitled “Efficacy of Submicron Dispersible Free Phytosterols on Non-2 Alcoholic Fatty Liver Disease: A pilot study” is well structured and presented. The main topics of the paper are well introduced, and the results and discussion are well organized.

The methods section must be clarified in respect to the advanced lipid profile analysis performed by Biosfer Teslab (Barcelona, Spain). A brief description of the method is important and references.

Another worrisome aspect is the absence of diet and lifestyle (exercise habits) control, despite the request to maintain their lifestyle and feeding habits. This limitation must be addressed in the discussion.

Author Response

REVIEWER 1:

Thank you for reviewing our research article ID: foods-2111692. Title: “Efficacy of Submicron Dispersible Free Phytosterols on Non-Alcoholic Fatty Liver Disease:  A pilot study” sent for consideration for publication in Foods.

Comment 1: The methods section must be clarified in respect to the advanced lipid profile analysis performed by Biosfer Teslab (Barcelona, Spain). A brief description of the method is important and references.

Answer: Thank you very much for this comment. A brief description of the advanced lipid profile analysis is provided and it is linked to a new reference.

New paragraph:

Page 3. Lines 116-124: “Liposcale®, lipoprotein profile characterization including lipid content (cholesterol and triglyceride concentration), size and particle number of the main lipoprotein classes, was performed by Biosfer Teslab (Reus, Spain). Serum samples were shipped on dry ice and were kept at −80°C until analyzed. Serum (200 ul) was diluted with deuterated water (50 µl) and 50 mM phosphate buffer solution at pH 7.4 (300 µl). Diffusion-ordered 1H NMR spectra were recorded at 306 K on a Bruker Avance III 600 spectrometer operating at a proton frequency of 600.20 MHz (14.1 T), as described in Mallol et al., 2015 [29].”

New reference:

Mallol, R.; Amigó, N.; Rodríguez, M.A.; Heras, M.; Vinaixa, M.; Plana, N.; Rock, E.; Ribalta, J.; Yanes, O.; Masana, L. Liposcale: a novel advanced lipoprotein test based on 2D diffusion-ordered 1H NMR spectroscopy [S]. J. Lipid Res. 2015, 56, 737-746.

Comment 2: Another worrisome aspect is the absence of diet and lifestyle (exercise habits) control, despite the request to maintain their lifestyle and feeding habits. This limitation must be addressed in the discussion.

Answer: Thanks to this observation, we included the following missing information in Methods, Results and Discussion sections regarding the control of lifestyle habits.

New paragraphs:

Page 2. Lines 82-85: “Audit and Short Form-36 Health Survey questionnaires were answered at the start and end of the study. Besides, lifestyle habits were controlled in visits every 3 months and in monthly telephone calls.”

Page 8. Lines 250-252: “There were no differences in alcohol consumption and quality life measures detected by Audit and SF36 questionnaires, respectively (data not shown). In control visits, patients referred no major changes to their lifestyle.”

Page 9: Lines 273-274: “Improvements were observed despite subjects maintained their basal low grade physical activity and general life style conditions.”

All changes introduced in the text are indicated in yellow and the references were renumbered.

Sincerely yours,

Prof. Rodrigo Valenzuela. PhD.

Associate Professor

Department of Nutrition

Faculty of Medicine

University of Chile

Reviewer 2 Report

Review on the manuscript "Efficacy of Submicron Dispersible Free Phytosterols on Non-2 Alcoholic Fatty Liver Disease: A pilot study" by Valenzuela et al..

The work is scientifically sound, and while there is new information here, the methodology novelty is somewhat adequate.

In a present state the manuscript submitted is adequate and recommend one minor correction. 

1. The work used a submicron, water dispersible free phytosterol (SDP) suspension to treat nonalcoholic fatty liver disease (NAFLD), therefore, more information about  the SDP suspension (such as photoes,  physicochemical properties ) should be provide in the revised manuscript.

Author Response

REVIEWER 2:

Thank you for reviewing our research article ID: foods-2111692. Title: “Efficacy of Submicron Dispersible Free Phytosterols on Non-Alcoholic Fatty Liver Disease:  A pilot study” sent for consideration for publication in Foods.

In a present state the manuscript submitted is adequate and recommend one minor correction.

Comment 1:  The work used a submicron, water dispersible free phytosterol (SDP) suspension to treat nonalcoholic fatty liver disease (NAFLD), therefore, more information about the SDP suspension (such as photoes, physicochemical properties) should be provide in the revised manuscript.

Answer: A physicochemical description of the dispersion (average particle size and pH) and a figure were included in Material and Methods and Supplementary Material, respectively.

Page 4. Lines 145-152: “SDP was a bench-scale dispersion manufactured by Nutrartis LLC, Santiago Chile. Each dose (10.5 mL) provides 2 g non-GMO, pine derived free phytosterols (70% sitosterol, 15% campesterol, 15% sitostanol, 5% stigmasterol) dispersed in water using 1.5% food grade surfactants (polysorbate 80 and oleic acid salts). It is a submicron aqueous dispersion with a density between 1 and 1.01 g/mL and pH between 8.5 and 10.5. Its particle size distribution, determined by dynamic light scattering, gave an average size of 300 nm, and percentiles 50, 75 and 90 corresponding to 270, 370 and 500 nm, respectively (Figure S1, Supplementary Material).”

 All changes introduced in the text are indicated in yellow and the references were renumbered.

Sincerely yours,

Prof. Rodrigo Valenzuela. PhD.

Associate Professor

Department of Nutrition

Faculty of Medicine

University of Chile

Reviewer 3 Report

Non-alcoholic fatty liver disease (NAFLD) is the most frequent liver pathology in the world, with a prevalence of 25% in general population. Intervention experimental research on non-alcoholic fatty liver disease is very meaningful.

The following content needs to be supplemented or modified

1, How was steatosis assessed? By MRI? Or lipid reduction? 

2, What does the 19% decrease in hepatic steatosis in the summary results refer to? Is the MRI showing a 19% reduction, or is it a 19% reduction in VLDL-TG?

3,There is no positive control in the research experiment, only the comparison before and after SDP intervention.

Authors need to revise the format.

4,The first time an abbreviation appears, the full name is required, such as line 75 CRO, line 97 GOT, GPT.

5,Figure 1 only has a legend, no figure. 

6, "2" should be superscripted in line 185 line 101, (BMI)< 17 Kg/m2 or > 40 Kg/m2.

Author Response

REWIEWER 3:

Thank you for reviewing our research article ID: foods-2111692. Title: “Efficacy of Submicron Dispersible Free Phytosterols on Non-Alcoholic Fatty Liver Disease:  A pilot study” sent for consideration for publication in Foods.

The following content needs to be supplemented or modified
Comment 1: 1, How was steatosis assessed? By MRI? Or lipid reduction?

Answer: In Page 3, Lines 93-94 of Inclusion Criteria it was already indicated that liver steatosis was assessed by Magnetic Resonance Imaging Proton Density Fat Fraction. Also, in Page 3, line 133 referred to Liver Steatosis Quantification where a brief description of the method was provided with references.

Comment 2: What does the 19% decrease in hepatic steatosis in the summary results refer to? Is the MRI showing a 19% reduction, or is it a 19% reduction in VLDL-TG?

Answer: In the Abstract (page 1, line 13), it was included, “magnetic resonance imaging”, for clarifying the method for liver fat measurement. In addition, on lines 16-17, it was clarified that “Liver steatosis relative change was -19%”.

Comment 3: There is no positive control in the research experiment, only the comparison before and after SDP intervention.

Answer: Indeed, this study was not compared to a positive control. For each parameter, the comparison between end point and base line were provided as relative change.

Comment 4: Authors need to revise the format.

Answer: Thank you very much for this comment. We carefully reviewed the format of the manuscript.

Comment 5: The first time an abbreviation appears, the full name is required, such as line 75 CRO, line 97 GOT, GPT.

Answer: CRO in line 75 was eliminated as it was the only time it was mentioned. But the name of the research organization contracted was kept in the text. Aspartate- (GOT) or Alanine- (GPT) aminotransferases was included in line 97.

Observation 6: Figure 1 only has a legend, no figure.

Answer: Thank you very much. Now it is included.

Comment 7:  "2" should be superscripted in line 185 line 101, (BMI)< 17 Kg/m2 or > 40 Kg/m2.

Answer: Both superscripts were included.

All changes introduced in the text are indicated in yellow and the references were renumbered.

Sincerely yours,

Prof. Rodrigo Valenzuela. PhD.

Associate Professor

Department of Nutrition

Faculty of Medicine

University of Chile